# Deep Mining: Detecting Anomalous Patterns in Neural Network Activations with Subset Scanning

## Abstract

This work views neural networks as data generating systems and applies anomalous pattern detection techniques on that data in order to detect when a network is processing a group of anomalous inputs. Detecting anomalies is a critical component for multiple machine learning problems including detecting the presence of adversarial noise added to inputs. More broadly, this work is a step towards giving neural networks the ability to detect *groups* of out-of-distribution samples. This work introduces "Subset Scanning" methods from the anomalous pattern detection domain to the task of detecting anomalous inputs to neural networks. Subset Scanning allows us to answer the question: "Which subset of inputs have larger-than-expected activations at which subset of nodes?" Framing the adversarial detection problem this way allows us to identify systematic patterns in the activation space that span *multiple* adversarially noised images. Such images are "weird together". Leveraging this common anomalous pattern we show increased detection power as the proportion of noised images increases in a test set. Detection power and accuracy results are provided for targeted adversarial noise added to CIFAR-10 images on a 20-layer ResNet using the Basic Iterative Method attack.

## 1 Introduction

The vast majority of data in the world can be thought of as created by unknown, and possibly complex, normal behavior of data generating systems. But what happens when data is generated by an alternative system instead? Fraudulent records, disease outbreaks, cancerous cells on pathology slides, or adversarial noised images are all examples of data that does not come from the original, normal system. These are the interesting data points worth studying. The goal of anomalous pattern detection is to quantify, detect, and characterize the data that are generated under these alternative systems. Furthermore, subset scanning extends these ideas to consider *groups* of data records that may only appear anomalous when viewed together (as a subset) due to the assumption that they were generated by the same alternative system. Neural networks may be viewed as one of these data generating systems. The activations are a source of high-dimensional data that can be mined to discover anomalous patterns. Mining activation data has implications for interpretable machine learning as well as more objective tasks such as detecting groups of out-of-distribution samples.

This paper addresses the question: "Which of the exponentially many subset of inputs (images) have higher-than-expected activations at which of the exponentially many subset of nodes in a hidden layer of a neural network?" We treat this scenario as a search problem with the goal of finding a "high-scoring" subset of images × nodes by efficiently maximizing nonparametric scan statistics in the activation space of neural networks.

The primary contribution of this work is to demonstrate that nonparametric scan statistics, efficiently optimized over node-activations × multiple inputs (images), are able to **quantify** the anomalousness of a subset of those inputs (images) into a real-valued "score". This definition of anomalousness is with respect to a set of clean "background" inputs (images) that are assumed to generate normal or expected patterns in the activation space of the network. Our method measures the deviance between the activations of a subset of inputs (images) under evaluation and the activations generated by the background inputs. The challenging aspect of measuring deviances in the activation space

of neural networks is dealing with high-dimensional data, on the order of the number of nodes in a hidden layer $\times$ the number of inputs (images) under consideration. Therefore, the measure of anomalousness must be effective in capturing systematic (yet potentially subtle) deviances in a high-dimensional subspace and be computationally tractable. Subset scanning meets both of these requirements (see Section 2).

The reward for addressing this difficult problem is an unsupervised, anomalous-input detector that can be applied to any input and to any type of neural network architecture. Neural networks universally rely on their activation space to encode the features of their inputs and therefore quantifying deviations from expected behavior in the activation space has broad appeal and potential beyond detecting anomalous patterns in groups of images. Furthermore, an additional output of subset scanning not fully explored in this paper is the subset of nodes at which the subset of inputs (images) had the higher-than-expected activations. These may be used to *characterize* the anomalous pattern that is affecting the inputs.

The second contribution of this work focuses on **detection** of targeted adversarial noise added to inputs in order to change the labels to a target class Szegedy et al. (2013); Goodfellow et al. (2014); Papernot & McDaniel (2016). Our critical insight to this problem is the ability to detect the presence of noise (i.e. an anomalous pattern) across *multiple* images simultaneously. This view is grounded by the idea that targeted attacks will create a subtle, but systematic, anomalous pattern of activations across multiple noised images. Therefore, during a realistic attack on a machine learning system, we expect a subset of the inputs to be anomalous *together* by sharing higher-than-expected activations at similar nodes. Empirical results show that detection power drastically increases when targeted images compose 8%-10% of the data under evaluation. Detection power is near 1 when the proportion reaches 14%. In summary, this is the first work to apply subset scanning techniques to data generated from neural networks in order to detect anomalous patterns of activations that span multiple inputs (images). To the best of our knowledge, this is the first topic to address adversarial noise detection by considering images as a group rather than individually.

## 2 SUBSET SCANNING

Subset scanning treats pattern detection as a search for the "most anomalous" subset of observations in the data where anomalousness is quantified by a scoring function, $F(S)$ (typically a log-likelihood ratio). Therefore, we wish to efficiently identify $S^* = \arg\max_S F(S)$ over all relevant subsets of the data $S$. Subset scanning has been shown to succeed where other heuristic approaches may fail (Neill, 2012). "Top-down" methods look for globally interesting patterns and then identifies sub-partitions to find smaller anomalous groups of records. These approaches may fail when the true anomaly is not evident from global aggregates. Similarly, "Bottom-up" methods look for individually anomalous data points and attempt to aggregate them into clusters. These methods may fail when the pattern is only evident by evaluating a group of data points collectively. Treating the detection problem as a subset scan has desirable statistical properties for maximizing detection power but the exhaustive search is infeasible for even moderately sized data sets. However, a large class of scoring functions satisfy the "Linear Time Subset Scanning" (LTSS) property which allows for exact, efficient maximization over all subsets of data without requiring an exhaustive search Neill (2012). The following sub-sections highlight a class of functions that satisfy LTSS and describe how the efficient maximization process works for scanning over activations.

### 2.1 NONPARAMETRIC SCAN STATISTICS

This work uses nonparametric scan statistics (NPSS) that have been used in other pattern detection methods Neill & Lingwall (2007); McFowland III et al. (2013); McFowland et al. (2018); Chen & Neill (2014). These scoring functions make no parametric assumptions about how activations are distributed at a node in a neural network. They do, however, require baseline or background data to inform their data distribution under the null hypothesis $H_0$ of no anomaly present.

The first step of NPSS is to compute empirical $p$-values for the evaluation input (e.g. images potentially affected with adversarial noise) by comparing it to the empirical baseline distribution generated from the background inputs that are "natural" inputs known to be free of an anomalous pattern. NPSS then searches for subsets of data $S$ in the evaluation inputs that contain the most evidence for

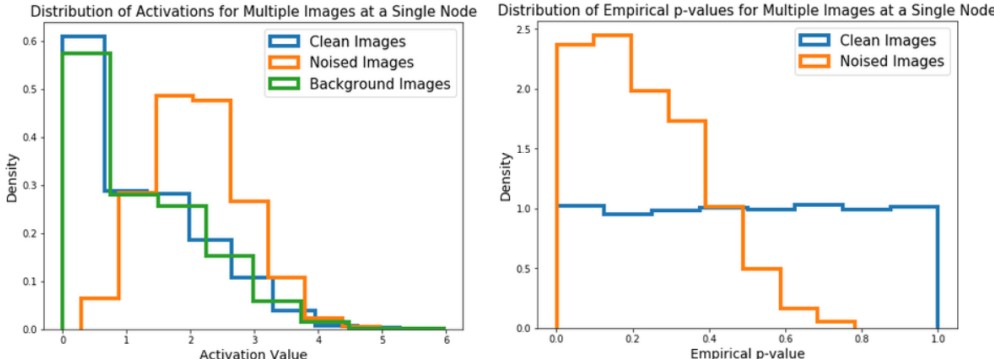

Figure 1: (a) Distribution of activations from three different image types at a single node under consideration. Noised images tend to have higher activations at this node than the background images. Clean images have activations similar to the background images. (b) Distribution of empirical $p$-values for clean and noised images at the same node considered in the left panel. Higher activations from noised images (as compared to the background) create larger number of smaller $p$-values. Activations from clean images (as compared to the background) create near-uniform distribution of $p$-values as expected under a null hypothesis.

*not* having been generated under $H_0$. This evidence is quantified by an unexpectedly large number of low empirical $p$-values generated by the evaluation inputs. See Figure 1.

In our specific context, we have evaluation data $D$ a collection of $N = |D|$ images $X_i$, which are fed through a neural network (defined by a set of nodes $O = \{O_1, \ldots, O_J\}$) producing an activation $A_{ij}$, at each node $O_j$. The challenge we face is detecting which subset $S = S_X \times S_O$, if any, of the evaluation images $S_X \subseteq D$ is anomalous as evidenced from their activation at some nodes $S_O \subseteq O$. To approach this challenge, NPSS uses as baseline data $D_{H_0}$ of known, clean CIFAR-10 images. Each of the $M = |D_{H_0}|$ background images $X_z$, also generates an activation $A_{zj}^{H_0}$ at each network node $O_j$.

Given $D$, our $N \times J$ matrix of activations for each evaluation image at each network node, and $D_{H_0}$, our corresponding $M \times J$ matrix of activations for each of the background images, we can obtain an empirical $p$-value for each $A_{ij}$: a means to measure of how anomalous the activation value of (potentially contaminated) image $X_i$ is at node $O_j$. This $p$-value $p_{ij}$ is the proportion of activations from the background images, $A_{zj}^{H_0}$, that are larger than the activation from the evaluation images $A_{ij}$ at node $O_j$. We note that McFowland III et al. (2013) extend this notion to $p$-value *ranges* such that $p_{ij}$ is uniformly distributed between $p_{ij}^{min}$ and $p_{ij}^{max}$. This current work makes a simplifying assumption here to only consider a range by its upper bound, $p_{ij}^{max} = \frac{\sum_{X_z \in D_{H_0}} I(A_{zj} >= A_{ij}) + 1}{M + 1}$.

The matrix of activations $A_{ij}$ is now converted into a matrix of $p$-values $P_{ij}$. Intuitively, if an evaluation image $X_i$ is "natural" (its activations are drawn from the same distribution as the baseline images) then few of the $p$-values generated by image $X_i$ across the network nodes–will be extreme. *The key assumption for subset scanning approaches is that under the alternative hypothesis of an anomaly present in the activation data then at least some subset of the activations, for the effected subset of images, will systematically appear extreme.* The goal is to identify this "high-scoring" subset through an efficient search procedure that maximizes a nonparametric scan statistic.

The matrix of $p$-values $P_{ij}$ from evaluation images is processed by a nonparametric scan statistic in order to identify the subset of evaluation images $S_X \subseteq D$ whose activations at some subset of nodes $S_O \subseteq O$ maximizes the scoring function $\max_{S = S_X \times S_O} F(S)$, where $S = S_X \times S_O$ represents a submatrix of $P_{ij}$, as this is the subset with the most statistical evidence for having been effected by an anomalous pattern.

The general form of the NPSS score function is $F(S) = \max_\alpha F_\alpha(S) = \max_\alpha \phi(\alpha, N_\alpha(S), N(S))$ where $N(S)$ represents the number of empirical $p$-values contained in subset $S$ and $N_\alpha(S)$ is the number of $p$-values less than (significance level) $\alpha$ contained in subset $S$.

This generalizes to a submatrix, $S = S_X \times S_O$, intuitively.

$N_\alpha(S) = \sum_{X_i \in S_X} \sum_{O_j \in S_O} I(p_{ij}^{max} < \alpha)$ and $N(S) = \sum_{X_i \in S_X} \sum_{O_j \in S_O} 1$.

There are well-known goodness-of-fit statistics that can be utilized in NPSS McFowland et al. (2018), the most popular is the Kolmogorov-Smirnov test Kolmogorov (1933). Another option is Higher-Criticism Donoho & Jin (2004). In this work we use the Berk-Jones test statistic Berk & Jones (1979): $\phi_{BJ}(\alpha, N_\alpha, N) = N * KL\left(\frac{N_\alpha}{N}, \alpha\right)$, where $KL$ is the Kullback-Liebler divergence $KL(x, y) = x \log \frac{x}{y} + (1 - x) \log \frac{1-x}{1-y}$ between the observed and expected proportions of significant $p$-values. Berk-Jones can be interpreted as the log-likelihood ratio for testing whether the $p$-values are uniformly distributed on $[0, 1]$ as compared to following a piece-wise constant step function alternative distribution, and has been shown to fulfill several optimality properties and has greater power than any weighted Kolmogorov statistic.

## 2.2 EFFICIENT MAXIMIZATION OF NPSS

Although NPSS provides a means to evaluate the anomalousness of a subset of node activations $S_O$ for a given subset of evaluation images $S_X$, discovering which of the $2^N \times 2^J$ possible subsets ($S = S_X \times S_O$) provides the most evidence of an anomalous pattern is computationally infeasible for moderately sized subsets of images and nodes. However, NPSS has been shown to satisfy the linear-time subset scanning (LTSS) property Neill (2012), which allows for an efficient and exact maximization over subsets of data.

For a pair of functions $F(S)$ and $G(X_i)$ representing the score of a given subset $S$ of data and the "priority" of a data record $X_i$ respectively, we have a guarantee that the subset maximizing the score will be one consisting only of the top-$k$ highest priority records, for some $k$ between 1 and $N$. If we consider a data record to be an image $X_i$, then our goal is to $\max_{S_X \subseteq \{S_1,...,S_N\}} F(S_X \times S_O)$, for a given subset of nodes $S_O$. The corresponding $G(X_i)$ function to measure the priority of an image, is the proportion of its $p$-values that are less than $\alpha$: $G_\alpha(X_i) = \frac{N_\alpha(X_i \times S_O)}{N(X_i \times S_O)}$. Therefore, from LTSS, we know that $\max_{S_X \subseteq \{X_1,...,X_N\}} F(S_X \times S_O) = \max_{\{X_{(1)},...,X_{(k)}\}_{k=1,...,N}} F(S_X \times S_O)$, where $X_{(k)}$ is the $k^{th}$ priority image.

Thus far we have described how to find the most anomalous subset of images *for a given subset of nodes*. Because $F(S)$ operates on a submatrix of $p$-value ranges, we can reorient the same process to identify an anomalous subset of nodes $S_O$ *for a given subset of images*. The goal is then to $\max_{S_O \subseteq \{O_1,...,O_J\}} F(S_X \times S_O)$, for a given subset of images $S_X$. The corresponding $G(O_j)$ function to measure the priority of an node, is the proportion of its $p$-value ranges that are less than $\alpha$: $G_\alpha(O_j) = \frac{N_\alpha(S_X \times O_j)}{N(S_X \times O_j)}$. Therefore, from LTSS, we know that $\max_{S_O \subseteq \{O_1,...,O_J\}} F(S_X \times S_O) = \max_{\{O_{(1)},...,O_{(k)}\}_{k=1,...,J}} F(S_X \times S_O)$, where $O_{(k)}$ is the $k^{th}$ priority node.

Given the two efficient optimization steps described above (optimizing over all subsets of images for a given subset of nodes, and optimizing over all subsets of nodes for a given subset of images), we are able to compute an efficient local maximum of $\max_{S_X \subseteq \{X_1,...,X_N\}, S_O \subseteq \{O_1,...,O_J\}} F(S_X \times S_O)$, via an iterative ascent procedure. To do so, we first choose a subset of attributes $S_O \subseteq \{O_1...O_J\}$ uniformly at random. We then iterate between the two LTSS-enabled optimization steps described above, until convergence, at which point we have reached a conditional maximum of the score function ($S_X$ is conditionally optimal given $S_O$, and $S_O$ is conditionally optimal given $S_X$). Moreover, we can perform multiple random restarts to approach the global optimum. LTSS enabled efficient optimization of NPSS has been shown to reach the global maximum with high probability empirically and theoretical conditions have been provided the guarantee exact identification of the truly affected subset of data McFowland III et al. (2013).

## 3 DETECTING ADVERSARIAL NOISE WITH SUBSET SCANNING

Machine Learning models are susceptible to adversarial perturbations of their input data that can cause the input to be misclassified Szegedy et al. (2013); Goodfellow et al. (2014); Kurakin et al. (2016a); Dalvi et al. (2004). There are a variety of methods to make neural networks more robust to adversarial noise. Some require retraining with altered loss functions so that adversarial images must

have a higher perturbation in order to be successful Papernot et al. (2015); Papernot & McDaniel (2016). Other detection methods rely on a supervised approach and treat the problem as classification by training on labeled noised examples Grosse et al. (2017); Gong et al. (2017); Huang et al. (2015). Another supervised approach is to use activations from hidden layers as features used by the detector. Metzen et al. (2017)

In contrast, our work treats the problem as anomalous pattern detection and operates in an unsupervised manner without apriori knowledge of the attack or labeled examples. We also do not rely on training data augmentation or specialized training techniques. Furthermore, our work is complimentary to many of the defenses mentioned above. For example, if one defense type requires the noising process to makes more extreme perturbations to change the class label, then those patterns should be more easily detected by subset scanning methods. A defense in Feinman et al. (2017) is more similar to our work. They build a kernel density estimate over background activations from the nodes in only the last hidden layer and report when an image falls in a low density part of the density estimate. This works well on MNIST, but performs poorly on CIFAR-10 Carlini & Wagner (2017). Our novel subset scanning approach looks at anomalousness at the node-level and across multiple inputs (images) simultaneously in order to detect patterns that span altered inputs (images).

### 3.1 TRAINING AND EXPERIMENT SETUP

We trained a ResNet20 (v1) residual neural network He et al. (2015) on 50,000 CIFAR-10 training images that had their mean pixel values subtracted. The test accuracy of the model was 0.9183. Future work will explore the effect of model classification accuracy on the ability to detect anomalous patterns within its activations. For this paper we chose Resnet20 for its relatively small size, popularity, and classification accuracy.

We focus our subset scanning methods on the final convolutional layer of the last residual block. This layer contains 64 filters each containing 8x8 nodes. Therefore our analysis is on the activations produced at these 4096 nodes. We do not use the spatial locations or filter membership of the nodes in the scanning process, however both of these could be useful extensions. For our adversarial experiments, we took $M = |D_{H_0}| = 9000$ of the 10000 validation images and used them to generate the background activation distribution ($D_{H_0}$) at each of the 4096 nodes. These images form our expectation of "normal' activation behavior for the network. The remaining 1000 images were used to form groups: "Clean" ($C$) and "Adversarial" ($A_t$) with Adversarial containing targeted noised versions of the 1000 images in Clean, repeated for each targeted class label $t \in (0 \ldots 9)$.

We used the Basic Iterative Method (BIM) adversarial attack Kurakin et al. (2016b) which is an improved version of the original Fast Gradient Sign Method Goodfellow et al. (2014). BIM has an $\epsilon$ parameter which controls how far a pixel is allowed to change from its original value when noise is being added to the image. We used a value of $\epsilon = 0.02$ in the scaled [0,1] pixel space. We also allowed the method to reach its final noised state over 10 steps with each of size 0.002. Smaller values of $\epsilon$ make the pattern subtler and harder to detect, but also less likely for the attacks to succeed in changing the class label to the target. All attacks were generated with the CleverHans package Papernot et al. (2016; 2018).

The images in $A_t$ were additionally reduced to remove any images that were *originally* classified as $t$ (and therefore the noise had no effect). The attacks had near 100% success rates. Target classes 1, 4, and 7 each had 1 failed attack that were also removed from their respective group, $A_t$. The end result is each $A_t$ group contains approximately 900 images that were successfully noised from an original predicted label *to* label $t$. The 1000 images in group $C$ are natural and have all class labels represented (nearly) equally.

We now proceed to generating tests sets that are composed of 500 images with varying percentages for clean images and adversarial images drawn randomly from group $C$ and $A_t$ without replacement, respectively. The percentages of adversarial images used in our test sets are 6%, 8%, 10%, 12%, and 14% for each of the target class labels. We additionally consider a special case where instead of drawing from a single group $A_t$, we uniformly draw from all $A_t$ groups. This is a particularly difficult test case because we do not expect there to be a common pattern consistent within images adversarialy targeted to separate classes. Each case is then repeated 100 times. Using $t = 0$ and $p = 0.10$ as an example, there would be 100 test sets of size 500 images composed of randomly selected 450 clean images (from all classes) and randomly selected 50 images noised to class 0.

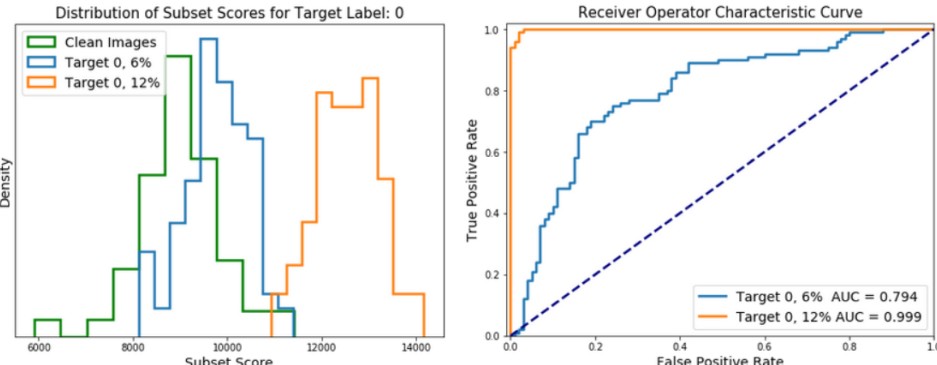

Figure 2: (a) Distribution of subset scores for test sets of 500 images. Test sets containing all natural images had lower than scores than test sets containing noised images. Higher proportion of noised images resulted in higher scores. (b) ROC curves for each of the noised cases as compared to the scores from test sets containing all natural images. The AUROC values for all cases are reported in Table 1.

The tests sets are scanned by our method in order to determine the highest-scoring (most anomlaous) subset of the 500 images. Each test set was evaluated with 5 random restarts where a random subset of nodes was generated as the seed for the iterative ascent. The most anomlaous subset returned by the scan is compared with the *true* state of the image (clean or noised). We then report *Detection Accuracy* as measured by precision and recall of the most anomalous subset. These results are shown in Table 1.

In order to measure *Detection Power* we must model the null hypothesis of no adversarial noise present. To do so we also create 100 test sets containing 500 clean images only; no noised images. Scanning these sets generates a distribution of scores under the null hypothesis. We then compare the distribution of scores generated when noised samples were present in the test set to the distribution of scores when scanning over a subset of 500 clean images. This analysis results in an ROC curve for each case of target $t$ and proportion $p$. The area under the ROC curve then measures detection power of our method for that particular case. Figure 2 demonstrates this process for target class 0 and proportions 6% and 12%. Detection Power results are reported in Table 1.

Finally, we consider experiments where images are scanned *individually* instead of as a larger group of 500 images. The process is identical to the one describe above except the test set size is 1 image instead of 500. We do not consider varying proportions in the individual case as each set is either $p = 0$ or $p = 1$. We also do not report precision or recall as there is no subset returned in this restricted case. We are able to report detection power by comparing the distribution of scores of an individual anomalous image to the distribution of scores created by scanning clean images individually.

## 3.2 RESULTS AND DISCUSSION

The top panel of Table 1 provides detection power for our experiments. Detection power is measured by Area-Under-ROC curves as demonstrated in Figure 2. The ability to detect targeted noise on *individual* images varies by class with moderate results and can be viewed as a performance floor. The focus of this work however, is the ability to detect adversarial noise across *multiple* images simultaneously. To that end, we show how detection power increases as the proportion of noised images increases in the test sets. At a proportion of 10% detection power is higher *as a group* than it is for an individual image across all target classes. Detection is nearly perfect for all classes at 12% and above. This suggests our scanning method is identifying a subtle anomalous pattern of activations that persists across multiple noised images targeting a single class.

We now focus on the "All" category which considers the test sets containing targeted examples from each of the 10 class labels. Detection power lags behind any single target class. This is because in the single target cases, our scanning method is exploiting an anomalous activation pattern that is consistent across multiple images. This pattern is less consistent when targeting different class

| BIM ($\epsilon = 0.02$) | Detection Power (AUROC) | | | | | |
|---|---|---|---|---|---|---|
| Target Class | Ind. | Proportion of noised images in the 500 image test set | | | | |
| | | 6% | 8% | 10% | 12% | 14% |
| 0 | 0.856 | 0.794 | 0.885 | 0.978 | 0.999 | 1.000 |
| 1 | 0.764 | 0.797 | 0.895 | 0.963 | 0.989 | 0.999 |
| 2 | 0.884 | 0.394 | 0.816 | 0.991 | 1.000 | 1.000 |
| 3 | 0.916 | 0.865 | 0.993 | 1.000 | 1.000 | 1.000 |
| 4 | 0.902 | 0.675 | 0.937 | 0.994 | 1.000 | 1.000 |
| 5 | 0.810 | 0.490 | 0.748 | 0.941 | 0.994 | 1.000 |
| 6 | 0.806 | 0.389 | 0.660 | 0.953 | 1.000 | 1.000 |
| 7 | 0.783 | 0.609 | 0.794 | 0.932 | 0.987 | 0.999 |
| 8 | 0.753 | 0.497 | 0.638 | 0.769 | 0.915 | 0.968 |
| 9 | 0.777 | 0.910 | 0.976 | 0.995 | 0.999 | 1.000 |
| All | – | 0.507 | 0.353 | 0.699 | 0.6044 | 0.769 |

| BIM ($\epsilon = 0.02$) | Detection Accuracy: Precision | | | | | Detection Accuracy: Recall | | | | |
|---|---|---|---|---|---|---|---|---|---|---|
| Target Class | Proportion of noised images in the 500 image test set | | | | | Proportion of noised images in the 500 image test set | | | | |
| | 6% | 8% | 10% | 12% | 14% | 6% | 8% | 10% | 12% | 14% |
| 0 | 0.196 | 0.275 | 0.364 | 0.494 | 0.620 | 0.828 | 0.903 | 0.923 | 0.941 | 0.946 |
| 1 | 0.216 | 0.286 | 0.352 | 0.415 | 0.480 | 0.800 | 0.831 | 0.862 | 0.866 | 0.875 |
| 2 | 0.228 | 0.363 | 0.638 | 0.794 | 0.852 | 0.894 | 0.950 | 0.971 | 0.973 | 0.974 |
| 3 | 0.267 | 0.470 | 0.618 | 0.726 | 0.772 | 0.947 | 0.969 | 0.966 | 0.966 | 0.962 |
| 4 | 0.232 | 0.341 | 0.505 | 0.642 | 0.746 | 0.889 | 0.920 | 0.941 | 0.940 | 0.941 |
| 5 | 0.196 | 0.290 | 0.401 | 0.541 | 0.654 | 0.836 | 0.899 | 0.913 | 0.935 | 0.944 |
| 6 | 0.180 | 0.349 | 0.561 | 0.673 | 0.725 | 0.733 | 0.896 | 0.955 | 0.963 | 0.964 |
| 7 | 0.200 | 0.271 | 0.358 | 0.442 | 0.519 | 0.822 | 0.845 | 0.878 | 0.894 | 0.896 |
| 8 | 0.166 | 0.238 | 0.299 | 0.367 | 0.436 | 0.678 | 0.763 | 0.803 | 0.848 | 0.869 |
| 9 | 0.256 | 0.342 | 0.410 | 0.477 | 0.527 | 0.845 | 0.860 | 0.882 | 0.878 | 0.886 |
| All | 0.109 | 0.160 | 0.240 | 0.255 | 0.337 | 0.403 | 0.484 | 0.625 | 0.566 | 0.702 |

Table 1: Detection Power and Accuracy for targeted adversarial noise added to CIFAR-10 images by the Basic Iterative Method attack. Results are provided for detecting individual images and subsets of 500 images where the number of noised images varies from 6% to 14%.

labels in the same test set. This suggests that targetted noise is activating the same set of nodes despite the original images coming from different classes.

In addition to Detection Power, Table 1 provides precision and recall measurements for the subsets of images identified by our scanning method. Precision is consistently lower than recall. We attribute this to two reasons. The first is that the 500 image test set contains targeted noised examples of a single class label, *as well as* natural images of that same class. Therefore, we believe the subset of anomalous images is likely to include the noised images *and* the natural images belonging to the target class, which decreases precision. Another reason for a relatively low precision is due to a static setting of a parameter to the scanning function, $\alpha_{max}$. For simplicity, this value was set to 0.5 for all runs and may be interpreted as assuming up to half of the data may be affected by the anomalous pattern. This is an inflated value which can be lowered if investigators had an apriori belief on the prevalence of the affected subsets in their data (i.e. the 6%-14% used in our experiments). Lowering this value would almost certainly increase precision (and lower recall).

We now consider recall measurements located in the bottom panel of Table 1. Recall is exceptionally high in our experiments. Similar to the argument for low precision, the high recall values are due to a large, static $\alpha_{max}$ value. A hyper-parameter search is feasible in supervised settings. Instead these experiments were conducted in an unsupervised form with $\alpha_{max}$ set arbitrarily at 0.5. We

now highlight a more subtle strength of our method with regards to recall. All things being equal, increasing the number of the noised images should *decrease* the recall rate as there are more noised images to miss. However, in almost all target classes we observe steady trend or an increase. This demonstrates subset scanning's innate adaptability by maintaining strong recall despite the number of noised images more than doubling (from 6% to 14%).

# 4 CONCLUSION

This work uses the Adversarial Noise domain as an effective narrative device to demonstrate that anomalous patterns in the activation space of neural networks can be efficiently *quantified and detected* across a subset of inputs (images).

The primary contribution of this work to the data mining and deep learning literature is a novel, unsupervised anomaly detector that can be applied to any pre-trained, off-the-shelf neural network model. The method is based on subset scanning which treats the detection problem as a search for the highest scoring (most anomalous) subset of node activations $\times$ inputs (images) as measured by nonparametric scan statistics. This is the first work to apply subset scanning methods to neural network activations and represents a novel contribution to both domains.

Nonparametric scan statistics applied to neural network activations operate on three levels of anomalousness. The first level is at a single activation generated by an input at a node. The anomalousness of this activation is quantified by its empirical $p$-value that reflects how large this activation is compared to a "background" of activations from known, natural images at the same node. Of course, not every input that has a large activation at this node is anomalous.

We therefore must consider the second level measured by NPSS: the anomalousness of a *subset* of activations for a single image (or equivalently, a single node). This level identifies the most anomalous subset of empirical $p$-values from a single image (or a single node). Despite the exponentially many subsets to consider, this optimization can be done exactly by only considering a linearly-many number of subsets of activations Neill (2012). An image (or node) that has a large number of small $p-$values is considered to be more anomalous. However, the large activations that make one image anomalous may occur at different nodes than an image that is equally anomalous with high activations at a different subet of nodes.

This consideration brings us to the third and highest level of anomalousness for NPSS applied to neural network activations: identifying a *subset* of inputs (images) that have higher-than-expected activations (i.e. large number of low empirical $p-$values) at a *subset* of nodes. This search procedure uses the same efficient optimization step to iteratively ascend between identifying the most anomalous subset of nodes (for a given, fixed subset of images) and identifying the most anomalous subset of images (for a given, fixed subset of nodes). In practice, this scanning method is able to identify a high scoring subset of images $\times$ nodes from a search space of 500 images and 4096 nodes in 3.8 seconds on average.

Efficient optimization is important in the large search space of neural network activations. However, this work also demonstrated that the subset identified by the scanning procedure is relevant for an anomalous pattern of interest. The second contribution of this paper is providing empirical results that subset scanning can detect the presence of targeted adversarial noise. Furthermore the detection power, precision, and recall increase when images with targeted noise are considered together as a group. To the best of our knowledge, this is the first work to consider adversarial noise detection across a group of images. Most adversarial noise defenses only provide detection results for individual images as they fail to scale to detecting at the group level. In practical settings, if a neural network is under a targeted attack there will be systematic differences across the affected images. Our method is capable of detecting these subtle, but systematic, patterns at node activations across multiple images.

We also highlight that the adversarial noise detection task was performed completely unsupervised and orthogonal to the original goal of the trained ResNet: to attain high classification accuracy. This suggests that subset scanning over neural network activations will be relevant in a broad range of neural network applications.

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

# A  APPENDIX

This appendix is meant to provide implementation details that are relevant to readers wishing to implement their own version of our experiments. Direct code is provided anonymously on Github `https://github.com/hikayifix/adversarialdetector`.

We also provide code examples and psuedo code here as well.

## A.1  CODE SNIPPETS

This paper has a focus on reproducible results and has used as many vanilla settings as possible.

Our resnet training was done in Keras and was taken directly from `https://github.com/keras-team/keras/blob/master/examples/cifar10_resnet.py` We had to change steps per epoch to

```python
# to model.fit_generator
steps_per_epoch = x_train.shape[0] / batch_size
```

Our noised attacks were generated by CleverHans and on the final 1000 images of the test set. The first 9000 were kept clean and used to form our baselines of expected activations under $H_0$.

```python
from cleverhans.attacks import BasicIterativeMethod
from cleverhans.utils_keras import KerasModelWrapper
##Basic Iterative Method
targs = np.array([[0,0,0,0,0,0,0,0,0,1]])
sess =  K.get_session()
wrap = KerasModelWrapper(model)
bim = BasicIterativeMethod(wrap, sess=sess)
bim_params = {'eps': 0.02,
              'nb_iter': 10,
              'eps_iter': 0.002,
              'y_target': targs}
adv_x = bim.generate_np(x_test[9000:10000,:,:,:], **bim_params)
```

Extracting activations from the network.

```python
import keras.backend as K
def get_layer(inp, out, layer_array):
    return K.function([layer_array.layers[inp].input],
      ↪ [layer_array.layers[out].output])
#Sample extraction --
f1_func = get_layer(0, 1, model)  #get acts from first layer
f1_act = f1_func([x_test])        #get acts from all images
```

The activations from the 9000 background images were sorted offline. Then at runtime we use np.searchsorted to determine where an evaluation activation would fall among the sorted background activations at each node. This was used to efficiently calculate $p$-value ranges.

## A.2 PSEUDOCODE

Inputs: evaluation dataset, validation dataset, training dataset, $\alpha_{\max}$, $Y$.

1. Train a Neural Network from the training dataset.

2. For each image $X_i$ and each Node $O_j$, in both validation and evaluation datasets, compute the activation $A_{ij}$ given the network.

3. Compute the $p$-value range $p_{ij} = [p_{\min}(p_{ij}), p_{\max}(p_{ij})]$ corresponding to each activation $A_{ij}$ in the evaluation dataset.

4. Iterate the following steps $Y$ times. Record the maximum value $F^*$ of $F(S)$, and the corresponding subsets of images $X^*$ and nodes $O^*$ over all such iterations:

   (a) Initialize $S_O \leftarrow$ random subset of nodes.

   (b) Repeat until convergence:

      i. Maximize $F(S) = \max_{\alpha \leq \alpha_{\max}} F_\alpha(S_X \times S_O)$ over subsets of images $S_X \subseteq \{X_1, \ldots, X_N\}$, for the current subset of nodes $S_O$, and set $S_X \leftarrow \arg\max_{S_X \subseteq \{X_1, \ldots, X_N\}} F(S_X \times S_O)$.

      ii. Maximize $F(S) = \max_{\alpha \leq \alpha_{\max}} F_\alpha(S_X \times S_O)$ over all subsets of nodes $S_O$, for the current subset of images $S_X$, and set $S_O \leftarrow \arg\max_{S_O \subseteq \{O_1, \ldots, O_J\}} F(S_X \times S_O)$.

5. Output $S^* = S_X^* \times S_O^*$ as well as $F(S^*)$.

