# OpenReview forum: "Deep Mining: Detecting Anomalous Patterns in Neural Network Activations with Subset Scanning"
_ICLR.cc/2020/Conference — Reject_

### Official Review · AnonReviewer1 · 2019-10-21
**Official Blind Review #1**

**Rating:** 3

**Review:**

The paper proposed a scheme to detect the presence of anomalous inputs, such as samples designed adversarially for deep learning tasks, that is based on a "subset scanning" approach to detect anomalous activations in the deep learning network. The paper is considering a very interesting problem and provides the suitable application of an approach previously developed for pattern detection. The approach is motivated by p-value statistics of the activation patterns in the deep learning network under the "null hypothesis" of a non-anomalous input.

My rating is "weak reject" because the explanation of the subset scanning approach is not clear. The paper describes two functionals F and G that are defined over subsets of the data and activations. Section 2.2 mentions a method that maximizes F over data and activation subsets by maximizing F over the data subsets under a fixed activation subset and vice versa, and iterating over these two. The section also discusses the function G that measures the priority of an image, but does not describe how the priority is known to provide an optimal solution for the former optimization over F. Another function with the same name is defined to measure the priority of an activation node, and again it is not clear why this will provide an optimal solution for the latter optimization. It would have been helpful to establish (or at least instantiate) the optimality results for this approach; only a citation is provided.

The applicability of the approach is limited to the requirement that multiple adversarial samples be present for accurate detection. It would appear that other approaches to anomaly detection do not require this condition, particularly if they are designed to operate on individual samples. Furthermore, the requirement for the "same system" to design the anomalous samples limits the applicability of the anomaly detection approach to cases like the single adversarial design detection setting studied here.

Some questions for the authors:
Is there a reason why the figures (Fig. 2) show results only for a single class?
Is there a reason why no other anomaly detection algorithms for the activation patterns were used in comparisons? How about other adversarial noise detection algorithms?
Is there intuition behind the difference in performance when all target classes vs. a single target class is used? Does this mean that a multi-class adversary generation would be too diverse for the proposed approach to detect it effectively?

Minor comments
"Weird together" - is too informal, and it's not clear why this phrasing is needed. Consider replacing or explaining.
Typos "anomlaous" multiple times.

**Experience Assessment:**

I have read many papers in this area.

**Review Assessment: Checking Correctness Of Derivations And Theory:**

I assessed the sensibility of the derivations and theory.

**Review Assessment: Checking Correctness Of Experiments:**

I assessed the sensibility of the experiments.

**Review Assessment: Thoroughness In Paper Reading:**

I read the paper at least twice and used my best judgement in assessing the paper.

---

> ### Author Response · Authors · 2019-11-14
> **Thanks!**
>
> Thank you for taking the time to review our work!
>
> Allow us to take some more page-space here in the comments with a longer prose form of the optimality of each individual maximization step.  (The convergence of these individual steps to a global maximum is addressed in a larger, top-level comment).
>
> Recall the function G as a priority function which ranks elements under consideration.  When optimizing over images for a fixed set of nodes the elements are the images.  The priority function is the number of node activations created by the image that are less than a threshold level alpha; the higher number of these nodes, the higher priority of the image.
>
> Now, lets consider the subset of images formed exactly by the 1st, 2nd, and 4th highest priority images.  (Note the 3rd priority image is missing).  The LTSS property allows us to guarantee that this subset (1st, 2nd, and 4th) is suboptimal.  This is because the score of this subset can be improved by either a) removing the 4th priority image form the subset or b) adding the 3rd priority image to the subset.  In other words,
>
> F(1,2,4) <= F(1,2) OR F(1,2,3,4)   (where 1,2,3,4 are the priority rankings of images provided by G).
>
> Therefore we know there are at most LINEARLY many subsets of images to consider and those subsets all have the form of the top-k priority images for some k between 1 and n.  Any subset not meeting this form does not need to be considered.  This drastic reduction of the search space allows us to score only the necessary subsets while guaranteeing that the highest scoring one will be one of them.
>
> (Another version of this same logic can be summarized as:   If the kth highest priority image is a part of the highest scoring subset then we know all images with a higher priority than the kth must also be included.)
>
> Proving that the scoring functions used in this current work satisfy the LTSS property is more involved and is shown on page 1541 here, http://www.jmlr.org/papers/volume14/mcfowland13a/mcfowland13a.pdf.  It comes down to the non-parametric scan statistics being monotonically increasing with the number of p-values less than a threshold alpha.
>
> re: Multiple samples being required.
>  The philosophy of subset scanning is that it is important to leverage a larger, group structure to identify a pattern that may not be noticeable in individual elements.  In the adversarial noise detection scenario, this boils down to caring more about detecting a coordinated attack on the system as compared to a more random (chaotic neutral) actor who may be motivated to alter a single image to a random new class (a single, un-targeted attack).
>
> However, we do not completely ignore the individual image detection power of the method.  These are reported in the first column on the large table of results and can be thought of as an effective floor for detection power when moving into group scanning.
>
> Adversarial noise is not the only scenario where multiple examples of a similar anomalous pattern may be present.  We've also considered the 'new class label' problem where a network may have been trained on cats and dogs, but now houses are becoming more frequent in the data stream.  Detecting a single house may be difficult, but detecting a dozen houses (among hundred+ cats and dogs) is more reasonable.  This type of formulation also has implications for detecting data-drift over time as well.  In these settings a single one-off anomalous example is not enough to declare a larger shift in the data being processed.
>
>
> Class 0 was simply chosen because it was the first class.  Those images are more intended to demonstrate to readers how we calculated AUC's coming from the scores generated by tests sets containing ALL clean images compared to the scores generated by evaluating tests sets that contain some group of noised images.
>
> We've added a top-level comment under this review for why many comparisons across networks, noise types, detection algorithms, data sets, etc. were left out.  The short summary is we believe the idea of detecting out-of-distribution samples by leveraging a common anomalous pattern that is present across those samples is more relevant to the larger body of work then another SOTA claim (which at the current rate of arxiv submissions is fleeting at best).
>
> The lower detection power when noised examples in the test set are coming from different classes is a noteworthy result.  This suggests that the power of detecting from a single class is truly leveraging a persistent pattern.  It is not simply doing a good job on individually anomalous images and then cleverly combining them.  If it was doing that, then detection power should also be high when the images had different targets.

---

### Official Review · AnonReviewer3 · 2019-10-24
**Official Blind Review #3**

**Rating:** 3

**Review:**

The authors apply linear time subset scanning to find groups of anomalous inputs and network activations. They further use this method to detect effects of the same adversarial perturbation algorithm over a set of images.
Major comments:
Overall this is quite interesting work, and seems like a promising method to detect groups of anomalies. However, this work is not complete without comparison to existing methods, the lack of which makes it impossible to evaluate its usefulness. See [1,2].

Minor questions/comments:
The results seem to be applied to ReLU activation networks and anomalous signals are described as having higher activation that the background or the clean images. Could you clear up whether this method works for all activations or just ReLU networks?
I’m envisioning a case where the anomalous images are all anomalous because they are all very similar. If I put 100 of the same (or very similar images) would this be something that this method could detect?
Your results seem to suggest that adversarial perturbation methods produce adversarial images with activations that are more extreme than natural images. This doesn’t seem immediately obvious to me and I would be interested in further exploration of this direction.
A value of \epsilon=0.02 was used in experiments for BIM. While I agree that smaller values of \epsilon are harder to detect it would be useful to have an evaluation of performance over smaller values of \epsilon even if these did not perform as well.
[1] Xiong, L., P ́oczos, B., Schneider, J., Connolly, A., VanderPlas, J.: Hierarchical probabilistic models for group anomaly detection. In: AISTATS 2011 (2011)
[2] Chalapathy R., Toth E., Chawla S. (2019) Group Anomaly Detection Using Deep Generative Models. In: ECML PKDD 2018. Lecture Notes in Computer Science, vol 11051. Springer, Cham

**Experience Assessment:**

I have published one or two papers in this area.

**Review Assessment: Checking Correctness Of Derivations And Theory:**

I carefully checked the derivations and theory.

**Review Assessment: Checking Correctness Of Experiments:**

I assessed the sensibility of the experiments.

**Review Assessment: Thoroughness In Paper Reading:**

I read the paper thoroughly.

---

> ### Author Response · Authors · 2019-11-14
> **Thanks!**
>
> Thank you for bringing this piece to our attention.  Chalapathy R., Toth E., Chawla S. (2019) Group Anomaly Detection Using Deep Generative Models.   A quick review does seem to show a similar philosophy in the goal of detecting patterns that persist across multiple images/inputs.   Future work will consider this in direct comparisons due to their ability to also scale to groups.
>
> ReLu is actually trickier than sigmoid and tanh in this regard and that is because it is difficult for an activation to be anomalously 'low' due to the large prevalence of '0' activations.  Therefore we only considered anomalously 'high' activations.  (The difference is simply defining a left or right tailed pvalues).   With sigmoid and tanh it is possible to define anomalous in either direction (i.e. 2 tailed pvalues).  However, we felt that relu seems to be the most common activation function and did not want give the impression that a user's  model would have to be retrained before being 'scannable'.
>
> We briefly considered scanning over 'pre-relu' activations which does have a notion of some activations being more negative than others.  This did not help detection performance.  We suspect its because in terms of propagating information through the network, a severely negative activation (pre-relu) is no different than a barely negative one (post-relu).  In short, scanning for larger than expected values on post-relu activations was the best balance for direct explanations of the overall process.
>
> Excellent question on performance whether an anomalous pattern would be detected by a dominant class of the same image type (and all clean examples).  We did not consider that.  Our clean images in the test sets were always constructed with a uniform sampling from the larger group of images.  This means that each class would've been equally represented.  We suspect the correct way to explore this would be through evaluating the role that alpha_max plays.  We currently have an agnostic role of alpha_max by leaving it at 0.5.  By decreasing this value we would be giving preference to a smaller number of nodes that produce more extreme (smaller pvalue) activations.   This parameterization would likely downplay the role of the scenario you described.  In fact, we currently estimate that an alpha_max value of 0.5 is too high.  A smaller value may potentially decrease recall, but drastically increase precision.  This type of question is easily addressed if we wish to go into supervised detection.  However, at that point the more interesting question is which layers provide the best detection power.
>
> Re: Noised examples producing more extreme activations than clean ones.  Other work in this space https://arxiv.org/pdf/1810.08676.pdf considers detection power over multiple layers of a smaller, simpler network.   Detection power varies across layers and, critically to your point, in some layers the noised examples produced much more 'boring' activations than clean ones.  This is noted by a AUC value << 0.5.  The authors of that piece propose that adversarial noise is running 'interference' by removing the signal normally present in clean images. Once that signal is removed, a new label assignment can be made trivially.    The current work under review only considers the final convolution layer for scanning.   It is possible that other layers provide higher detection results, or as shown in the arxiv piece, some layers may appear 'hollow' for noised images.

---

### Official Review · AnonReviewer2 · 2019-10-27
**Official Blind Review #2**

**Rating:** 6

**Review:**

The paper is the first paper, in my knowledge, that introduces the problem of identifying anomalous (or corrupted) subset of data input to a neural network. The corrupted inputs are identified vis-a-vis a set of “clean” background set (e.g., the training/ validation data set). Experimental evaluation is performed on the problem of identifying the subset of noisy CIFAR-10 images created by adding targeted adversarial perturbations.

To achieve this, the problem is posed as that of subset detection and subset scanning approaches are adapted for the same. The activation of a node under consideration is converted to a p-value (how extreme the activation is, for that input data, w.r.t. the background set).

A goodness of fit is then defined for any subset using the Berk-Jones test statistic which is used to create an NPSS scoring function to be maximized over the set of all subsets of the product set – (set of nodes x set of input images). The authors suggest that this combinatorial problem can be addressed efficiently as NPSS has been shown before to have the linear-time subset scanning (LTSS) property (Neill 2012).

However, since the search is now over a product subspace and the LTSS property holds for individual subspaces (selection of activation nodes or input images while the other is held fixed), the LTSS property doesn’t trivially extend to the product space. The authors suggest an algorithm which alternatively iterates between the two LTSS steps. Since the above algorithm is not guaranteed to obtain the global optimum, multiple random restarts are suggested. This is the weakest part of the paper – neither is a formal proof of convergence provided, nor is the efficiency of the propped algorithm demonstrated empirically. Also, the performance of the algorithm (accuracy) can only be weakly used to judge the goodness of the local optimum obtained as it confounds the power of NPSS with the gap between the local and global optimum.

Contributions

In summary, the main contribution of the paper is the introduction of the problem of the identification of anomalous subsets of adversarially corrupted examples on deep neural networks and the first approach to address the problem via a NPSS scoring function that (partially) satisfies the LTSS property suggesting the existence of efficient (and exact?) algorithms for solving the problem. The work has incremental novelty as it seems to be largely based on prior art (not on DNNs). Initial results look promising.

Negatives

However, I have the following concerns:

- (Clarity) Mathematical formulation for the NPSS score function and the optimization problem to be solved should be clearly stated (and not inside paragraphs of text).

- The proposed algorithm is ad hoc - an obvious extension of the standard algorithm that exploits the LTSS property to the product of two set spaces. The algorithm may not find a good optimum. No proof, analysis or study of its properties – optimality gap, convergence, and, efficiency is presented.

- At deployment, the attacks may be non-targeted or the target may be different for different corrupted inputs. This will dilute the signature as the set of nodes where the ‘extreme values’ appear may not be the same across the corrupted examples. It is not clear if it is the case if the target is the same. In this case, subset scanning may not work. This assumption should’ve been directly tested.

- ‘Berk-Jones can be interpreted as the log-likelihood ratio for testing whether p-values are uniformly distributed on [0,1] vs. … alternate distribution’. Why should this be the test? P-values on clean, background data can be directly obtained and whatever that distribution can be empirically tested or uniformity can be tested against after a whitening transformation?

- Comparison with reasonable ‘baselines’ – for example, with Feinman et al. 2017? Comparison with current state of the art defenses on CIFAR10?

- Why is the BIM attack (Kurakin et al. 2016b) used? There are more powerful attacks available now?

- The NPSS scoring function is supposed to maximize over significance values \alpha. What range was used and how was it decided? In the discussion on Detection Power reported in Table 1, it is suggested that different \alpha_max will trade off precision vs recall. This has not been experimented with.


**Experience Assessment:**

I have read many papers in this area.

**Review Assessment: Checking Correctness Of Derivations And Theory:**

I assessed the sensibility of the derivations and theory.

**Review Assessment: Checking Correctness Of Experiments:**

I assessed the sensibility of the experiments.

**Review Assessment: Thoroughness In Paper Reading:**

I read the paper thoroughly.

---

> ### Author Response · Authors · 2019-11-15
> **Thanks!**
>
> Thanks for your detailed feedback and taking the time to understand our work!
>
> We've left a larger, top-level comment regarding convergence of the iterative algorithm.
>
> re: targeted attacks for different labels
> This work was done and can be found in the last row of each of the 3 tables labeled "ALL".  In these test sets, the anomalous (adversarially noised) images were drawn uniformly from ALL target labels.  The 10 rows preceding the ALL rows have their noised images drawn only from one particular class.
>
> We note the lower AUC, precision and recall under this "ALL" setting.  What this highlights, is that the search algorithm is actually leveraging commonality across multiple images that all have the same targeted class.  We know this, because when we take that commonality away, the performance drops.   This implies the group-detection power is not working simply because there are more than 1 noised example image in the test set but because those noised examples share a common, anomalous pattern in the activation space of the neural network.
>
> An interesting debate can occur between whether it is more important to detect a random-chaotic actor attempting to noise random images to random target as compared to detecting a malicious actor who wants to change specific images to a specific class.  Our proposed method is obviously designed for the latter, more interesting scenario.
>
> re: New alternatives to Berk-jones.
> Thanks for the idea(s).  New scoring functions that operates on sets of data records that satisfy the linear-time subset scanning (LTSS) property are being found/proved regularly.  It is quite possible that a scoring function that compares the observed pvalues of a test image to the observed pvalues of known-to-be-clean images could also satisfy LTSS.  We did not consider extensions in that direction (yet).  We may find out that p-values coming from clean images are not actually uniform in practice.
>
> We believe a larger (and more interesting) concern of the use of these non-parametric scoring functions is their assumption of independence between activations.  That is, under the null hypothesis of no anomaly present in the data, each activation is assumed to be generated independently from the others.  This assumption is ok when you are optimizing over images (because each activation comes from a separate image), however, it is highly violated when optimizing over nodes because those activations are coming from the same image.  We know the activations in the network processing the same image will be correlated.  What this means from a practical perspective is the anomalous scores of a clean images are likely higher than they should be and this hurts detection power.  We believe future work in correcting for this independence assumption will yield better returns. (Note, it is ok for activations to be correlated under the alternative hypothesis).
>
> re: Other attacks;  Please see our top-level comment on this query.
>
> re: Alpha max
>
> The range used was between 0.0 and 0.5 (technically 0.0 was replaced with whatever the smallest observed pvalue was).  0.5 was chosen simply as the midpoint between 0 and 1.  This follows a recommendation laid out in subset scanning literature http://www.jmlr.org/papers/v14/mcfowland13a.html.
>
> In a supervised setting, we would encourage tuning alpha_max in order to improve detection power and accuracy.  As mentioned in the text, a smaller alpha max would give preference to a smaller number of imagesXnodes that have more extreme pvalues.  A larger alpha max allows for a larger number of imagesXnodes to be returned that are all slightly anomalous.   However, if an investigator wishes to prioritize smaller subsets of more extreme observations, we would recommend the 'Higher Criticism' statistic. (David Donoho and Jiashun Jin.  Higher criticism for detecting sparse heterogeneous mixtures.Annals of Statistics, 32(3):962–994, 2004.)   This HC scan statistic naturally gives preference to smaller, more extreme subsets as compared to Berk-Jones and Kolmogorov Smirnov which both prefer identifying deviances in the 'middle' of a distribution and not the tails.
>
> Of course varying both alpha-max and choice of scan statistics takes up page space as well as alluding to a supervised detection problem.  We wished to keep the results section clear on focusing on  how detection power/accuracy changes as the proportion of anomalous images (containing the same anomalous pattern) increases.

---

### Author Response · Authors · 2019-11-14
**Convergence to global max**

We will try and create a single thread for each topic that appeared in multiple reviews.  Therefore, multiple reviewers can respond to this topic-based thread (as opposed to multiple threads per reviewer).

Thanks.

Convergence
We can prove that the optimization algorithm will converge (perhaps to a local max) because there are a large but ultimately finite number of subsets to evaluate.  Second, at each iteration of the algorithm we only update the current subset if the new one has a higher score.  The scores will never decrease.  Combining these two points mean the algorithm will eventually converge.  This type of logic is the same used to show simpler iterative ascent algorithms (such as k-means) also converge

As for finding the global optimum, this is tricky and is part of (longstanding) on-going work in the subset scanning field.  For a bit more detailed background we reference a JMLR article that uses the same iterative ascent algorithm.  http://www.jmlr.org/papers/volume14/mcfowland13a/mcfowland13a.pdf
(this piece is cited many times in our submission).

They empirically address convegence to a global max by defining an "approximation ratio".  This ratio is the % of the time that the converged score is at least 100 - % away from the true global max.  (They identify the true global max by an exhaustive search).

Empirically, they arrived at an approximation ratio of 100%. This means, out of their 100 attempts the algorithm found the highest scoring joint subset everytime.  (We believe that was done with 50 random restarts on each of the 100 trials).  An unconstrained version of their work had a ratio of 98%.

Now, a very large caveat of that comparison to our work is that it was done on a maximum of 16 attributes (also those attributes were arranged in a bayesian network and not a resnet as in our work).  Larger values of that were computationally infeasible for the exhaustive method to identify the true global max.  That problem is multiple orders of magnitude larger in our context where the number of nodes (i.e. features) we are considering are in the 10's of thousands.  This of course rules out attempts at exhaustively finding the global max for comparable "approximation ratio" calculations.

Some empirical work done by us (not included in the paper) shows the distribution of scores attained by each individual restart.  The high score attained among these restarts is reached by several restarts.  This implies there is some agreement on the scores they reached.  We have yet to find a case where the maximum score of all restarts was achieved by only 1 restart.  Of course, the plural of anecdote is not data.  Furthermore, even if there is large agreement of the same high score attained by multiple restarts, we have no way of knowing if that value is the true global max.

We now discuss the practical implications of the algorithm failing to identify the global max.  A key question is whether we believe the convergence differs under null and alternative hypotheses.  If convergence to the true global optimum is more likely under the alternative (where some subset of images have higher activations at the same subset of nodes) than the null (where no such images were intended to have that behavior), then our detection results would be biased higher.  One could claim this as a feature, not a bug ;).  In the other scenario, our detection results would be biased lower.

In either case, we claim that increasing the number of restarts should increase the probability of finding the global maximum under both null and alternative hypotheses.  Therefore, an appropriate follow-up analysis of this work for an appendix piece would be analyzing detection power as a function of increasing the number of restarts used in each optimization.  If a strong direction is observed in detection power then we would know which of the two scenarios described above may be true.  However, in practice we will always recommend the most number of restarts as limited by (parallel) compute power.

---

### Author Response · Authors · 2019-11-14
**Where are comparisons with method X on architecture Y using dataset Z?**

We will try and create a single thread for each topic that appeared in multiple reviews.  Therefore, multiple reviewers can respond to this topic-based thread (as opposed to multiple threads per reviewer).

Thanks.

Other Attack Types, Other detection methods, Other architectures, Other image data sets.

Thank you for the consistent feedback across all the reviewers on these points.  It is well noted (and even expected).

Throughout the process of exploring how to apply anomalous pattern detection techniques to neural network activations we have considered the following situations (and most of these in some form of product space)

Architectures: LeNet, Resnet20, Resnet56V2, Densenet
Data Sets: MNIST, CIFAR, FashionMnist, SVHN
Attacks: FGSM, BIM, Carlini-Wagner, DeepFool  (We are also considering new class labels and out-of-distribution detection)
Detection: ODIN, Mahalanobis Detector
Layers:  Shallow, Deeper, Avg Pooling, etc.

When it comes time to summarize results, one could quickly get overwhelmed. In order to clearly state the take-home message of this submitted work (which all reviewers correctly understood), we had to make severe cuts to the overall space of experiments.

We now re-emphasize that take home message:  Images that have targeted adversarial noise added to them are easier to detect when considered as part of a group.  This suggests the targeted noise is manipulating the activations of the network in similar ways across different images.

We will  now try and demonstrate that we are not cherry-picking results by showing a relatively small number of the overall experiments that were run.  In lieu of reviewing a 20+page appendix, we will summarize some of our observations not shown in the paper.

DenseNet and Resnet had similar results.  We used Resnet due to more familiarity.

ResnetV2 (with batch normalization) drastically increases detection power of this method when considering individual images (one at a time).  In fact, individual detection is so high (mid-to upper 90's as compared to mid 80's shown on this submission based on Resnet20v1), it was difficult to claim that the GROUP scanning added much benefit.  Therefore the insight that targetted noise manipulates the network in similar ways across multiple samples is lost.

If the goal was to claim SOTA detection results, we would push more in that space and emphasize the role that batch normalization played in increasing detection power with subset scanning.  It would also make more sense to compare to existing detection methods because the results would be based on an image-by-image detection metric.

However, this work's goal was to emphasize that out-of-distribution detection CAN be done at both a subset of nodes level as well as a subset of images level.  We believe these ideas will stand the test of time longer than fleeting SOTA detection power claims (plus it is a matter of time before Carlini breaks any individual image detection algorithms...).

Other work in this space considered optimizing over subset of nodes, but not a subset of images.  That piece was limited to a smaller network based on LeNet.  However, it did consider Carlini Wagner attacks.  As expected, CW attacks were harder to detect than BIM.  However, the interesting result of that work is how the detection power drastically varies over the layers.

Finally, we note strong detection results reported in a Neurips paper last year. https://arxiv.org/pdf/1807.03888.pdf  A direct comparison to this method (which also builds on ODIN and Feinman's artifacts work) is difficult because how they train their detector in a supervised method with labeled examples of the attack (or OOD samples).  The work under consideratoin here is done UNsupervised without labeled examples of the positive classes. Furthermore, they are not capable of applying their metric to a GROUP of images.  This latter point was the key reason why those detection comparisons were left out.  We want to reiterate (multiple times) that the main of idea of this current work is to demonstrate that detecting anomalous patterns across multiple images is possible and relevant.

We conclude by noting that an interesting research direction would be to combine the ideas presented in the Mahalanobis detector https://arxiv.org/pdf/1807.03888.pdf with the ideas of subset scanning presented here.  However, it is not trivial to allow methods designed to score individual images to now scale to groups of images.

---

### Decision · Program_Chairs · 2019-12-19

**Decision:**

Reject

**Comment:**

The paper investigates the use of the subset scanning to the detection of anomalous patterns in the input to a neural network. The paper has received mixed reviews (one positive and two negatives). The reviewers agree that the idea is interesting, has novelty, and is worth investigating. At the same time they raise issues about the clarity and the lack of comparisons with baselines. Despite a very detailed rebuttal, both of the negative reviewers still feel that addressing their concerns through paper revision would be needed for acceptance.